# Failure Analysis of a Flare Tip Used in Offshore Production Platform in Qatar

**DOI:** 10.3390/ma13153426

**Published:** 2020-08-03

**Authors:** Elsadig Mahdi, Ali Esmaeili

**Affiliations:** Department of Mechanical and Industrial Engineering, College of Engineering, Qatar University, 2713 Doha, Qatar; esmaeili.64@gmail.com

**Keywords:** flare, failure, Incoloy 800H, tensile strength, microhardness

## Abstract

An immature failure of a gas flare tip used in Qatar oil and gas offshore industry was investigated throughout this study. The design lifetime of the flare was fifteen years; however, it manifested immature failure resulting in a reduction of its lifetime to ten years. The flare is composed of different parts where the upper flare body and wind deflector showed failure while other components were still healthy. The material used for the aforementioned failed parts was Incoloy 800H, which is a highly corrosion and high-temperature resistant steel alloy. The material was rolled up and welded together with different welding joints. The root cause of failure was identified by using chemical analysis and microstructural and mechanical characterizations. For the mechanical characterization, an optical microscope (OM) and scanning electron microscope (SEM) equipped with energy dispersive spectroscopy (EDS) analyses were used for the specimen extracted from the failed part in order to ensure that the material mentioned by the manufacturer demonstrated the same metallurgical properties. For the mechanical characterization, two sets of specimens were used, one close to the failure region and the other far from the failure area. The chemical analysis revealed that the material was truthfully Incoloy 800H. The mechanical examination results showed a significant reduction of mechanical properties, i.e., the ultimate tensile strength (UTS) and microhardness dropped by 44% and 41% for samples close and far from the failure regions, respectively. Careful examination of the failed parts indicated that failure mostly took place in the vicinity of the welds, in particular near the joints. Improper joint designs, as well as a number of joints being designed in tiny areas, worsened the harmful effect of the heat-affected zone (HAZ), resulting in crack nucleation in the HAZ regions. The effect of welding in a combination of harsh service conditions of flare caused further crack extension where they merged, resulting in final immature failure.

## 1. Introduction

The oil and gas industry is accompanied by the generation of a significant amount of gas emissions and high temperatures. It is well known that high-temperature service conditions affect the materials used [1,2,3,4,5,6]. The gas flaring system is an example of this severe condition. The gas flaring system’s primary purpose is pressure relief [7,8,9,10,11,12,13,14,15,16,17]. Flares are considered as the last defense line in plants that produce flammable products. One of their primary duties is to burn harmful, unneeded gases safely, and light liquids into the atmosphere. This paper examines the failure of the flare tip, which is generally made from stainless steel or Incoloy 800H. The flare tip service condition is harsh, which accelerates the corrosion process. Failure Analysis (FA) has been widely used as a tool to probe the root failure of a failed component using comprehensive data collection and analyses and to prevent future immature failure along with increasing the lifetime and safety of an industrial system [18]. A failure incidence can take place as a consequence of a malfunctioning interaction of materials, load and design, which can finally appear as any change in shape, dimension and mechanical property of an industrial component [19]. Excess hydrocarbon gases safely disposed of and burnt off in an eco-friendly way released extra gases; catastrophic failure may be caused by overpressure in other components of a petrochemical plant. Thus, a flaring system plays a critical role in refinery industries. In other words, the safe burning off of extra gases whenever needed should be continued in service without any shut down [20,21]. Typically, a flaring system is composed of other subcomponents, including a flare stack and a piping system that collects unwanted gases to be burnt off, as shown in Figure 1. 

The flare-tip at the end of the stack is designed to assist inlet air into the flare resulting in more efficient burning (Figure 2). 

The molecular seals fixed in the stack hamper flashback of the flame while a vessel installed at the base of the stack eliminates and preserves any liquids from the unwanted gas to the flare [22]. The material used for the flare tip is generally high-temperature oxidation resistant. In other words, the material used for the flare tip should possess high resistance against flame impingement, which generally takes place at low rates of flared gas. It is worth noting that a majority of failures in flare tips result from low relief gas rates because the tip undergoes the harmful impact of flame impingement [20]. In a study done by Yousefi et al. [23], the root failure of a gas flare used in the Iranian oil and gas industry was investigated. According to their results, the existence of sour gases, hydrocarbon gases and carbon dioxide arisen from the combustion processes along with the high service temperature approximately between 500 °C and 900 °C resulted in surface oxidation and carburization of the flare tip. As a result, the oxide layer appearing on the surface of the flare body as well as surface damage were the two main reasons for fertilizing the crack nucleation in the vicinity of the welding regions. Generally, the flare body is made up of a rolled sheet, while welding is used to join the end interface. The welding itself causes the weldments in the flare body to be inclined to crack initiation. Thus, immature failure that is expected in the welding regions can be attributed to the harsh and highly corrosive environment, along with the high temperature of the flare [24]. Ihsan et al. [25] studied the performance of advanced coating on gas flare systems made of 310 stainless steel (SS310). They found that using coatings with the double-layer thermal barrier increases the lifetime of the flare system. Failure analysis of some components of a flare gas system was investigated in a case study performed by Elshawesh et al. [26]. Based on their results, the failure took in expansion joints called bellows as subcomponents of the flaring system, which were made of Incoloy 625. The internal surface of the failure components was exposed to the highly harsh environment, including exposure to CO_2_ and H_2_S at 60 °C and 40 MPa pressure while the external surface exposed to winds, salt spray and sand storm. Finally, they stated that fatigue crack initiations were the root cause of failure resulting from the curvature design of the bellows itself, i.e., stress concentration, the high chloride content of the bellows surface, high service temperature along with the alternating vibration, i.e., fatigue. A case failure analysis of a flare gas used in an offshore plant in Qatar was investigated throughout this study (Figure 2a). The design pressure and temperature were 3.8 bar and −15 °C to 185 °C. Non-Destructive Examination (NDE) was carried out before flare usage (based on the company datasheet), including X-ray for 10% of all butt welds as well as the joint. The joint was related to the joints between the flare body and wind deflectors. Moreover, the penetration test (PT) was performed for all welds to examine surface cracking on the flare component surfaces. According to the site data, the flare was exposed to a highly aggressive environment, i.e., a high temperature of 700 °C and high humidity. Moreover, the structure showed immature failure, typically in the vicinity of the welding region, as shown in Figure 2b. According to the company datasheet, the design lifetime of the used flare gas was fifteen years. However, it demonstrated immature failure, a maximum lifetime of 10 years. 

## 2. Experimental Investigation Program

This section introduces the experimental investigation program carried out to identify the root cause of failure of the flare tip used in the offshore production platform in Qatar. The experimental investigation techniques include visualization, chemical analysis and microstructural and mechanical characterizations. To this end, samples taken from locations near and far from the observed cracks were examined. As shown in Figure 3, the failure mostly took place in the flare upper body part and wind deflector. 

Moreover, the failure region in the parts mentioned above is located in the vicinity of the welding regions, probably the heat-affected zone (HAZ). Since the material for the upper body and wind deflector are Incoloy 800H, two different samples were extracted from the flare for the failure analysis, i.e., one close to the failure and the other quite far from the failure. This was done to truthfully compare the mechanical properties of the material near and far from a failure, i.e., the consequence of the failure on final mechanical properties was to be examined. For the mechanical test characterization, tensile and hardness tests (Instron, Norwood, MA, USA) were performed on the two specimens. Figure 4 shows the tensile specimen prepared according to the ASTM E8M standard [27].

Five specimens were prepared for the tensile test, three specimens for a region near to the cracks and two specimens for a region far from the cracks. A crosshead speed of 2 mm/min was used for the tensile test. Brinell hardness test (HB) was also conducted at at least ten different points on each specimen. Table 1 shows the materials used for different components of the flare.

For the metallurgical investigation, the specimens were prepared according to ASTM E3-01 [28]. A metallographic examination was also carried out on the specimen extracted near the failure region in order to see the grain size and the twining system mechanism, which is a typical deformation mechanism for Incoloy 800H. In this context, first, the specimen was mounted, followed by polishing its surface using sandpaper, diamond paste range from 15 µm to 5 µm. Finally, the polished specimen was etched using Marble reagent (100 mL H_2_O + 50 mL HCl + 10 mL CuSO_4_) for 1 min at room temperature. Scanning Electron Microscope (Quanta 200 Environmental Scanning Electron Microscope (ESEM) with EDAX-EDS; Watertown, MA, USA) (SEM) and Optical Microscope (Leica DM25; Wetzlar, Germany) (OM) were used for the microstructural investigation. The chemical composition of the samples was first estimated using SEM equipped with Energy Dispersive Spectroscopy (EDS), while the possible phases and intermetallic compounds were also analyzed using X-Ray Diffraction (Malvern Panalytical; Malvern, UK) (XRD). Finally, the fracture surface of the specimen was analyzed using the secondary mode of SEM. 

## 3. Results and Discussion

### 3.1. Chemical Analysis and Microstructural Characterization

Figure 5 shows the chemical composition of the Incoloy 800H based on the standard and the analysis performed on the samples in this study. The chemical composition of the flare upper body and wind deflector was analyzed using EDS, and the results are compared with the chemical composition of the Incoloy 800H (based on the standard) for comparison. From the EDS analysis, the main elements of Incoloy 800H, i.e., Fe, Cr and Ni, can be detected, as shown in Figure 5, which is in accordance with the weight concentrations based on the standard. It is worth noting that the presence of other elements such as Al, Ti and C can also be detected by EDS analysis, as shown in Figure 5, though their weight concentration cannot be identified correctly using EDS due to the semi-quantitative analysis of EDS. Meanwhile, from the EDS analysis, the presence of oxygen can be attributed to surface oxidation of the materials, which might be one of the possibilities of failure. 

Figure 6 presents the microstructure of the Incoloy 800H. From the grain size and the twining systems, the material used for the flare upper body and wind deflector can be proved to be Incoloy 800H. The average grain size of 100 µm can be seen using the mean linear intercept methods. The microstructure of the material contains mainly the austenitic phase, i.e., Fe–Cr–Ni, which again demonstrates that material use is correctly Incoloy 800H. In addition, as can be seen from the SEM image highlighted by yellow arrows, the presence of precipitates is also common in the microstructure where the rich elements in the precipitates are Ti, N and C using EDS analysis. It can be concluded that the possible irregular precipitates dispersed in the microstructure are TiC.

Figure 7 shows the XRD analysis performed on the samples. Not only was the presence of main phases probed, including Ti, Cr and Ni, but some other intermetallic compounds such as FeNi, Al0.3Cr0.7 and CrNi3 can also be traced from the XRD results. By taking into account the microstructural characterizations made in this study, it can be concluded that the material used in this study truthfully manifested the same microstructural properties as the reference material. Thus, the possible reason for the failure might be related to the operation condition of the flare as well as an improper selection of the materials used for flare parts, which will be discussed later.

### 3.2. Mechanical Properties 

For the mechanical characterization, as mentioned before, two samples representing the materials extracted near and far from the failure region were selected in order to truthfully compare the possible impact of the failure on final mechanical properties. Figure 8 demonstrates the tensile strength of the samples. 

Specimens 3 to 5 represent the tensile behavior of the materials far from the failure, whereas specimens 1 to 2 demonstrate the tensile strength of the samples extracted near the failure. From Figure 8, it can be concluded that the Ultimate Tensile Strength (UTS) of the latter, i.e., the materials near the failure parts, substantially decrease, with up to 44% reduction in UTS. Moreover, the yielding strength also shows a significant reduction from almost 600 MPa to 200 MPa for the area far from the cracks and the area near the cracks, respectively. On the other hand, the materials selected close to the failure shows more significant elongation for the one extracted far from the failure, though both samples were cut from the sample plate. This clearly indicates that the mechanical properties of the material dramatically reduced in the vicinity of the failure as a consequence of the failure. 

Figure 9 shows macroscopic and microscopic images of the fracture surface of the tensile specimens. 

Specimens 1a,b and 2a,b represent the specimens far and near from the failure region, respectively. From the macroscopic image (Figure 9a,b), it is clear that the specimen near the failure (i.e., specimens 2a,b) manifest more considerable extension, i.e., higher elongation compared with the specimen taken far from the failure, i.e., specimen 1a,b. This again indicates the detrimental impacts of failure on the degradation of mechanical properties. In addition, as shown in Figure 9b, fracture took place at 45°, demonstrating the activation of the slip mechanism, which is again a typical fracture mechanism for a ductile material. Thus, the shear stress accounts for final failure for both specimens. SEM images of the fracture surface for specimens 1 and 2 are shown in Figure 9c–f, respectively. The white dashed circles represent the region where a higher magnification image was taken. The fracture surface of both specimens presents a typical fracture pattern of ductile materials where dimple morphology can be identified, i.e., a large amount of plastic deformation can be seen for both specimens. This can also be proved by the formation of microvoids along the fracture surface. The presence of inclusion accounts for void nucleation, as can be seen by some inclusion at the bottom of each dimple (Figure 9e,f). In addition, specimen 1 manifests higher surface roughness (Figure 9c,d) with respect to specimen 2 (Figure 9e,f), which can be attributed to the higher tensile strength of the former compared with the latter. A Brinell hardness test (HB) was also performed for both specimens to compare the consequence of failure on hardness properties. Specimens 1 and 2 represent the samples taken near the failure part, whereas specimens 3 and 4 extracted far from the failure region. Similar to tensile test results, the hardness of the area in the vicinity of the failure is smaller than the area far from it, as shown in Figure 10. The average hardness for the specimen close to the failure region is approximately 60 HNB, while the average microhardness for the specimen collected far from the failure region is 85 HNB, i.e., the microhardness dropped to 41%. This finding again clearly indicates that the base material used for the upper flare body, as well as wind deflector (Incoloy 800H), loses its mechanical properties during flare operation. 

### 3.3. The Root Cause of Failure

As discussed in the previous section, tensile strength and microhardness of the samples in the vicinity of the failure region substantially decreased, whereas the mechanical properties of the specimen cut far from the failure region demonstrated no change, i.e., showed the same mechanical properties like the primary material. As mentioned before, the failure mostly took place in the vicinity of the welding region for the flare upper body part and wind deflector, which were made of Incoloy 800H (Figure 3). This means that the welding procedure performed to join the parts might affect the mechanical properties, in particular, the presence of Heat Affected Zone (HAZ), which is the weakest region in welded structures [29]. Not only the welding procedure but also the improper design for welding joints, especially for joint as well as using a highly professional welder, might be other reasons for the failure that occurred in the vicinity of the welded areas. Another important fact that can be detected after the initial examination of the whole flare structure is that the failure mostly took place in the upper part of the flare body, whereas the lower part showed no failure. It is worth noting that the flare upper body parts are mainly made of Incoloy 800H, whilst the lower body part is made of SS316 (Table 1). This clearly indicates that, due to the harsh service condition of the flare, including 700 °C and 3.8 bar, the materials selected for upper body parts (Incoloy 800H) might not be chosen appropriately based on the real service condition of the flare. Based on these findings, it can be concluded that AISI 316 stainless steel is more applicable for high-temperature service than Incoloy 800H. Finally, as shown in Figure 3, the surface of the flare, in particular upper parts, severely corroded, which might be attributed to the sour gas (sulfurization), high service temperature and high humidity (high-temperature oxidation). As mentioned in the previous section, the materials used for flare upper body parts and wind deflectors are correctly Incoloy 800H based on the microstructural examination performed in this case study. Thus, the root cause for the failure should be found either in service conductions of the flare or improper welding design and implementation of welding done by the welder. After careful investigation of the flare components, the possible root cause of the failure can be attributed to improper design for welding connections, especially for the joints, which connected the wind defectors to the flare body. The flare upper and lower body parts were correctly welded using circumferential and longitudinal welds, as shown by black arrows in Figure 11. 

Thus, the welding performed to join the rolled Incoloy 800H plates to make them into cylindrical shapes was well done in terms of an appropriate welder, welding fillers and the proper welding procedure such as the number of welding paths, and possible heat treatment upon welding. On the other hand, many failure clues can be found mainly in the vicinity of the joint, as shown in Figure 12. This clearly indicates that the joint’s design is detrimental in terms of mechanical properties. The most critical place that failure took place is shown by black arrows in Figure 12a,b. As can be seen, the three joints were designed in a tiny area. This improper design caused the weakness of HAZ for all three joints accumulated in a tiny area. This can also be approved by the crack nucleation started in the vicinity of the joints for all three joins, i.e., it begun from the HAZ, which manifested the lowest mechanical properties for all welded materials. A combination effect of this HAZ along with the severe service condition of the flare itself, i.e., high temperature in the range of 700 °C as well as high humidity, worsens the failure scenario. Thus, the initiated cracks propagate gradually (white dashed lines) during flare service until they merge, followed by final catastrophic failure. Moreover, the conic shape of the flare that was composed of a circumference welding to join the conic part to the cylindrical part (See Figure 11) is another reason that intensifies failure due to the stress concentration resulting from the conic shape of the flare. It is worth noting that the conic shape itself is not a concern; however, when the joint is placed in the vicinity of it, it will act as a suitable place for stress concentration resulting in final failure. As shown in Figure 12c,h, the presence of minor failure in other parts of the flare is noticeable, mostly near the joints, as well as the holes in their vicinity. This observation can be seen in Figure 12f when the crack started from the joints and grew toward the holes. In this case, it seems that the quality of weld performed by the welder is not good. Thus the crack is initiated at the HAZ. Another minor failure that took place for the flare is related to the joint between the pockets and flare body (to hold the wind deflector) where a single joint was used for the connection. This improper design results in failure for most of the joints used for pocket [30].

## 4. Conclusion

The gas flare demonstrated immature failure mostly in two different pats, i.e., upper body part and wind deflector, which was made of Incoloy 800H. The design lifetime for the flare was fifteen years; however, it dropped to 10 years. Thus, chemical analysis and microstructural and mechanical examinations were carried out to find out the root cause of failure. In summary, the following outcome is derived from this study:The microstructure illustrated that flare material contained Fe, Cr and Ni, as well as the presence of the small amount of other alloying elements such as Al, Ti and C, which were in accordance with the reference material.The average grain size of 100 µm was obtained for the analyzed material, which was also in agreement with the reference materials.Presences of the twining system, as well as the formation of tiny precipitates such as TiC along the examined surfaces, were also noticed from the microstructural characterization.Careful examination of the failed parts indicated that failure mostly took place in the vicinity of the welds, in particular near the joints.Improper joint designs, as well as designing a high number of joints in tiny areas, worsened the harmful effect of HAZ, resulting in crack nucleation in the HAZ regions.

## 5. Recommendations

According to the discussion made in the previous sections, the following recommendations can be suggested:The configuration of the joints should be optimized in a way to reduce the harmful effect of HAZ.In addition, the number of joints should also be reduced due to the fact that they connect to the flare boy using a joint.Moreover, the blind designs for the joints used for pockets should be improved, i.e., at least a two-sides joint should be performed. Herein, finite element analysis will be very useful to avoid the inappropriate design of the joints.

## Figures and Tables

**Figure 1 materials-13-03426-f001:**
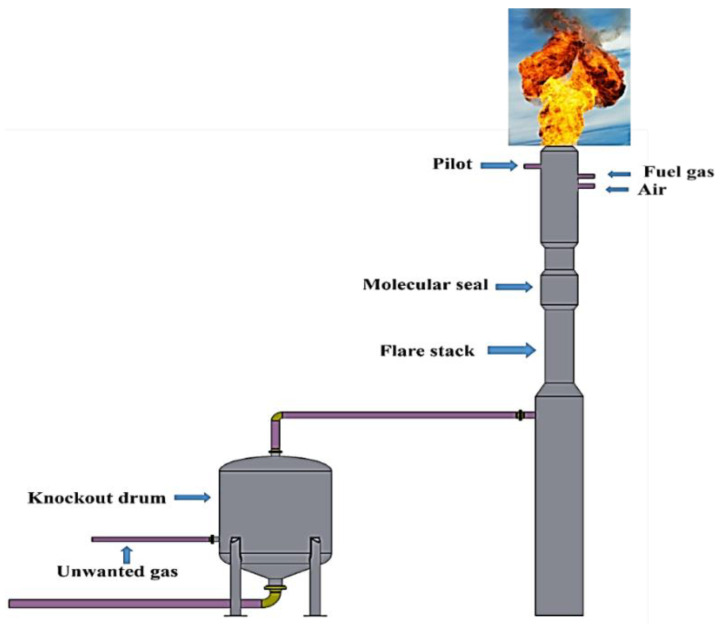
Flaring system component.

**Figure 2 materials-13-03426-f002:**
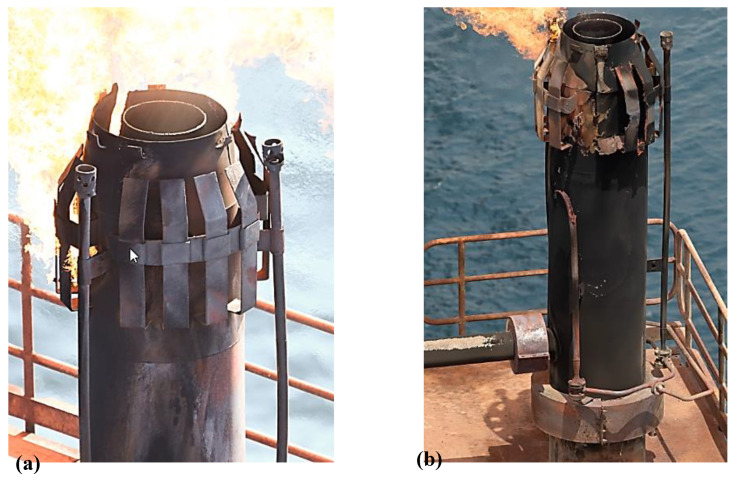
The investigated Gas flare Tip (**a**) Gas fare tip (**b**) Gas flare tip with immature failure.

**Figure 3 materials-13-03426-f003:**
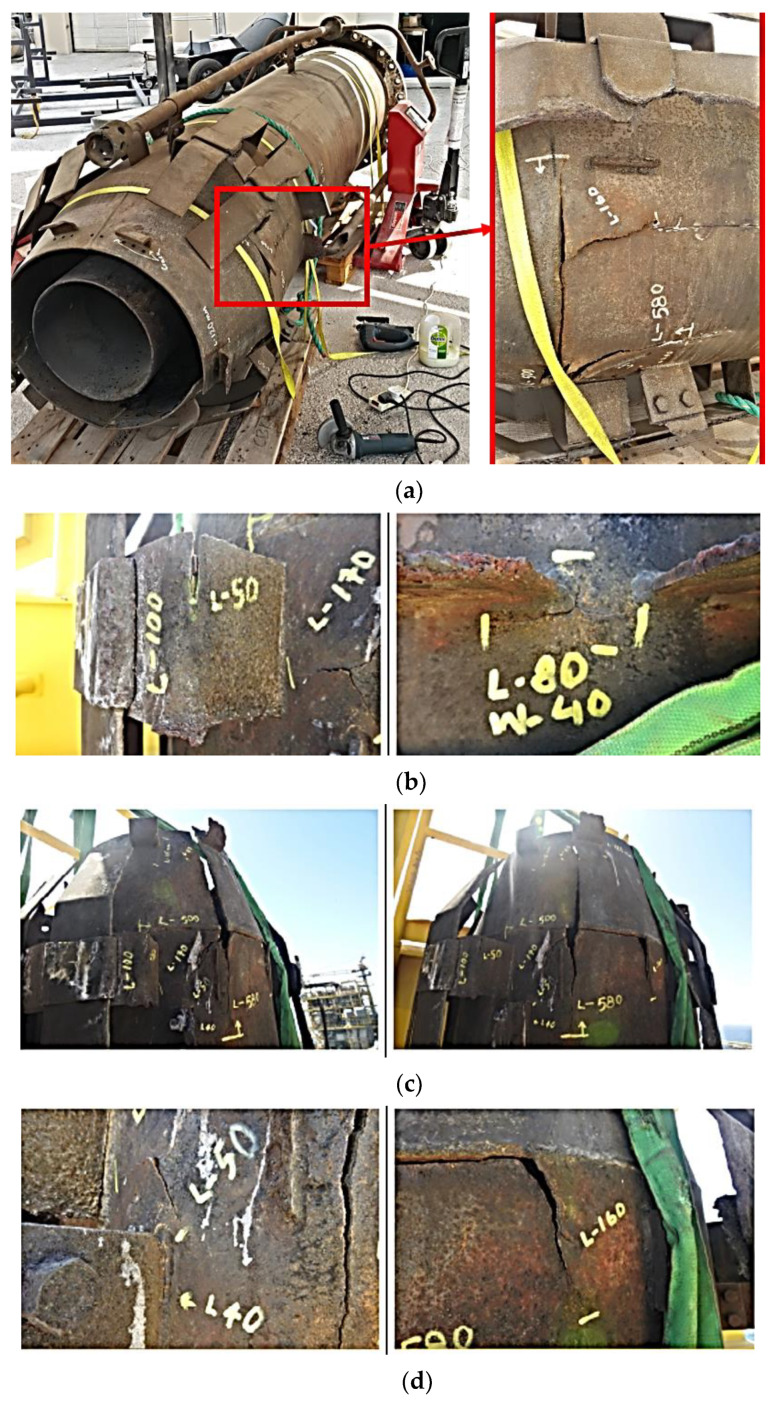
Failure regions of the different components of the flare in the vicinity of the welding region. (**a**) The cracks location. (**b**) Crack at wind deflector and crack on upper flare body near degraded wind deflector. (**c**) Crack at conical flare body. (**d**) Zoomed crack at conical flare body.

**Figure 4 materials-13-03426-f004:**
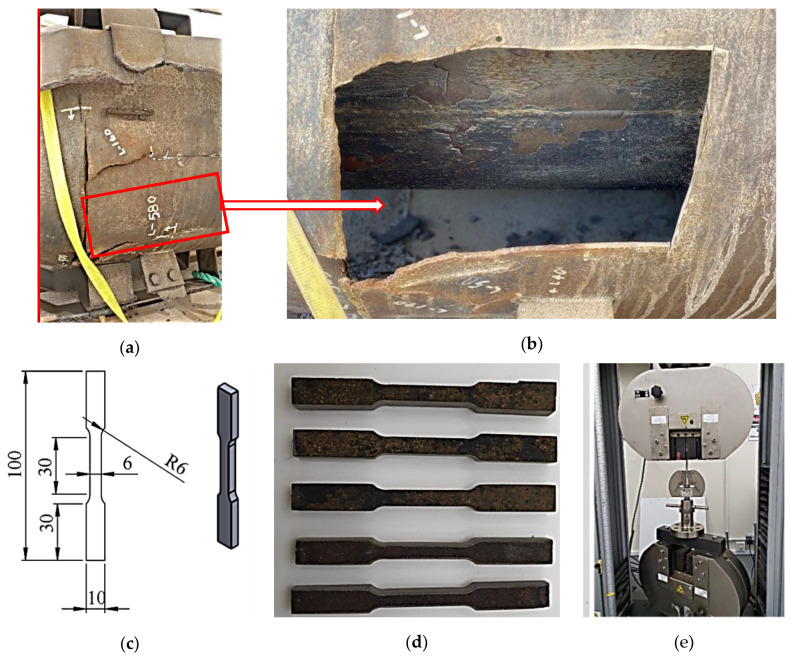
Specimen preparation for tensile test: (**a**) The flare tip region where the plate was cut. (**b**) The flare tip after the plate was removed. (**c**) The dimensions of the tensile test specimen. (**d**) The machined tensile test specimens. (**e**) Tensile test setup.

**Figure 5 materials-13-03426-f005:**
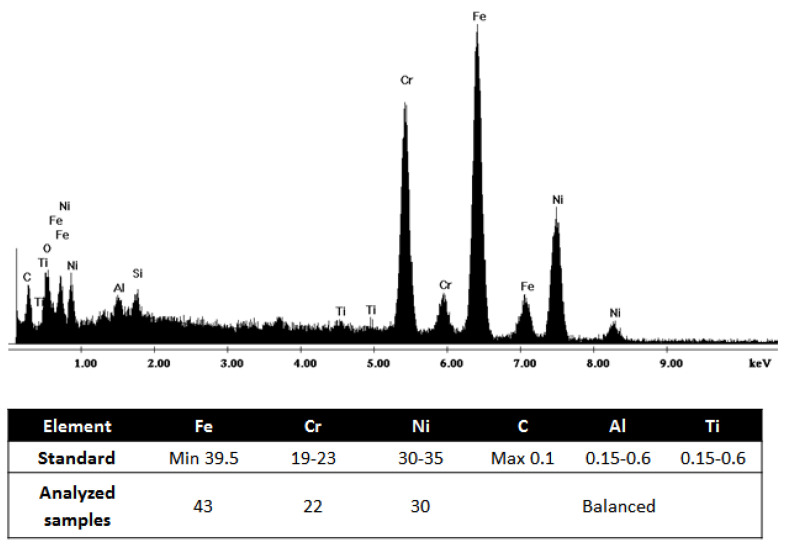
Chemical composition of the reference material as well as the samples using EDS analysis.

**Figure 6 materials-13-03426-f006:**
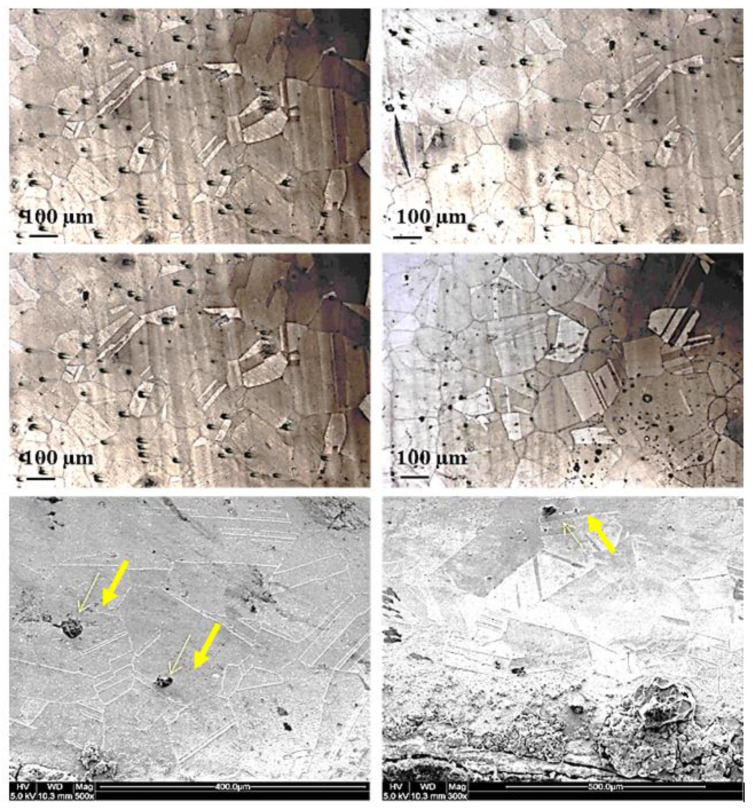
The microstructure of Incoloy 800H, the yellow arrows show the twinning system as well as the presence of precipitates.

**Figure 7 materials-13-03426-f007:**
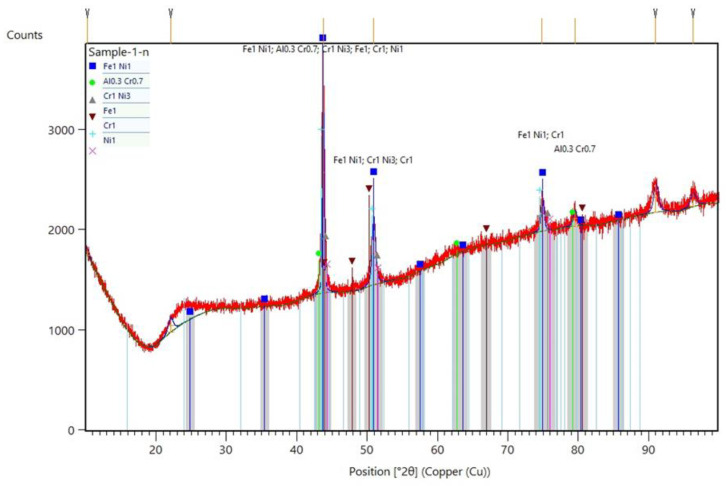
XRD analysis of the samples, demonstrating the presence of individual main elements including Fe, Cr, Ni and the intermetallic compounds.

**Figure 8 materials-13-03426-f008:**
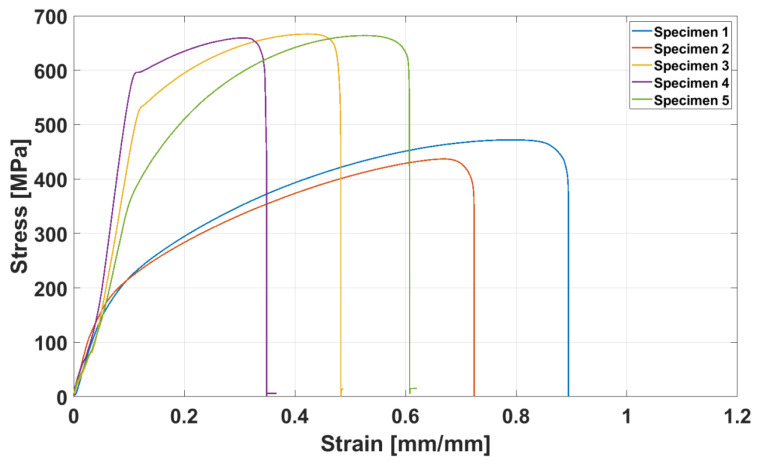
Tensile stress–strain curves.

**Figure 9 materials-13-03426-f009:**
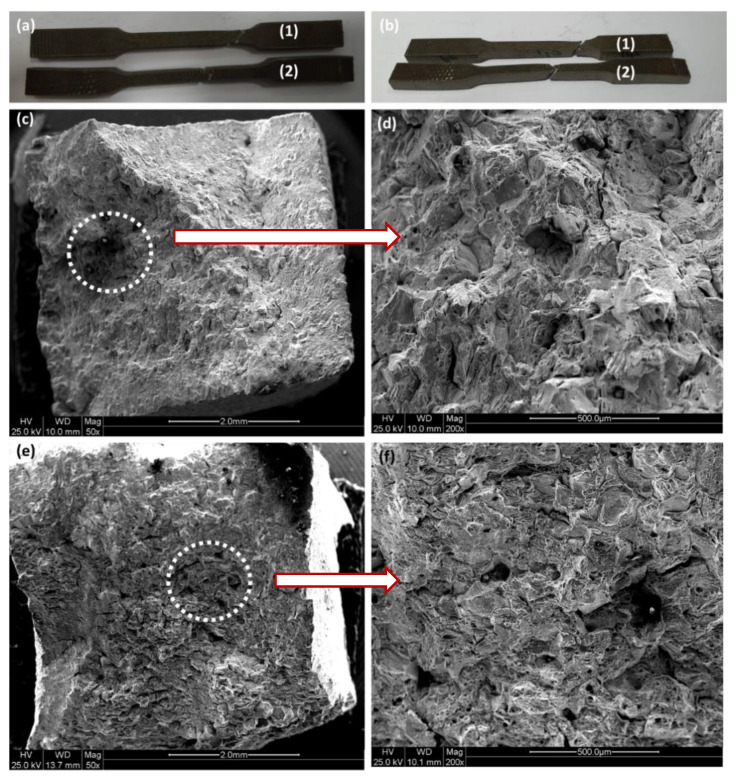
Fracture surface morphology: (**a**,**b**) Macroscopic images for specimens 1 and 2, respectively; (**c**,**d**) SEM images for specimen 1; (**e**,**f**) SEM images for specimen 2.

**Figure 10 materials-13-03426-f010:**
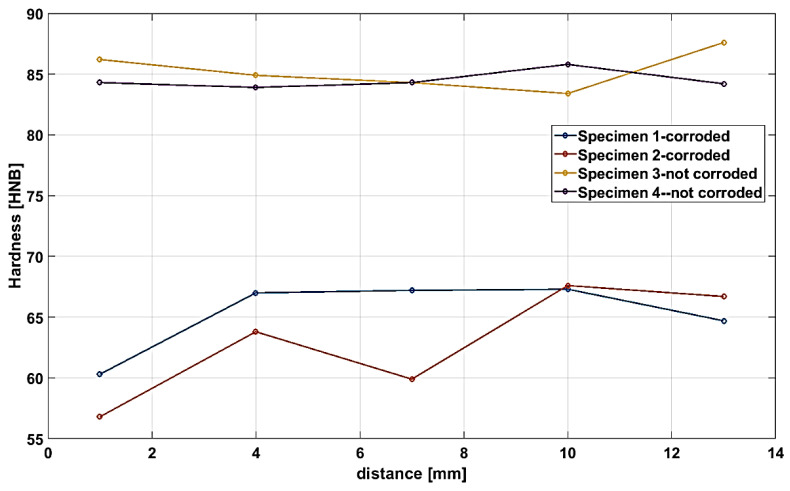
Microhardness of the specimens close and far from the failure region.

**Figure 11 materials-13-03426-f011:**
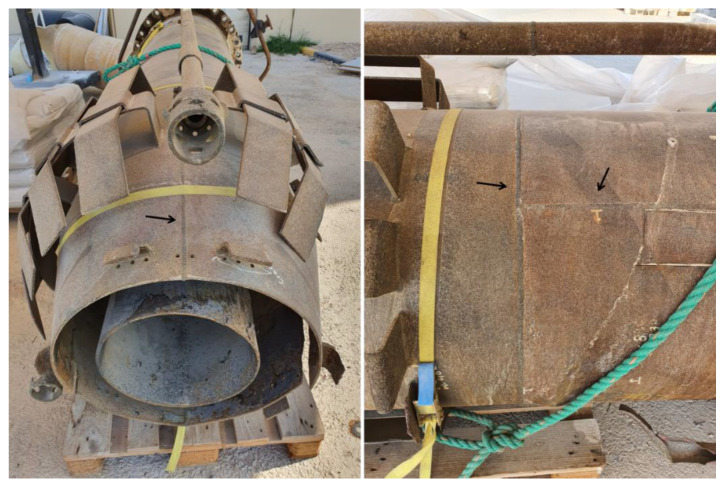
Longitudinal and circumferential welding line in the flare body part.

**Figure 12 materials-13-03426-f012:**
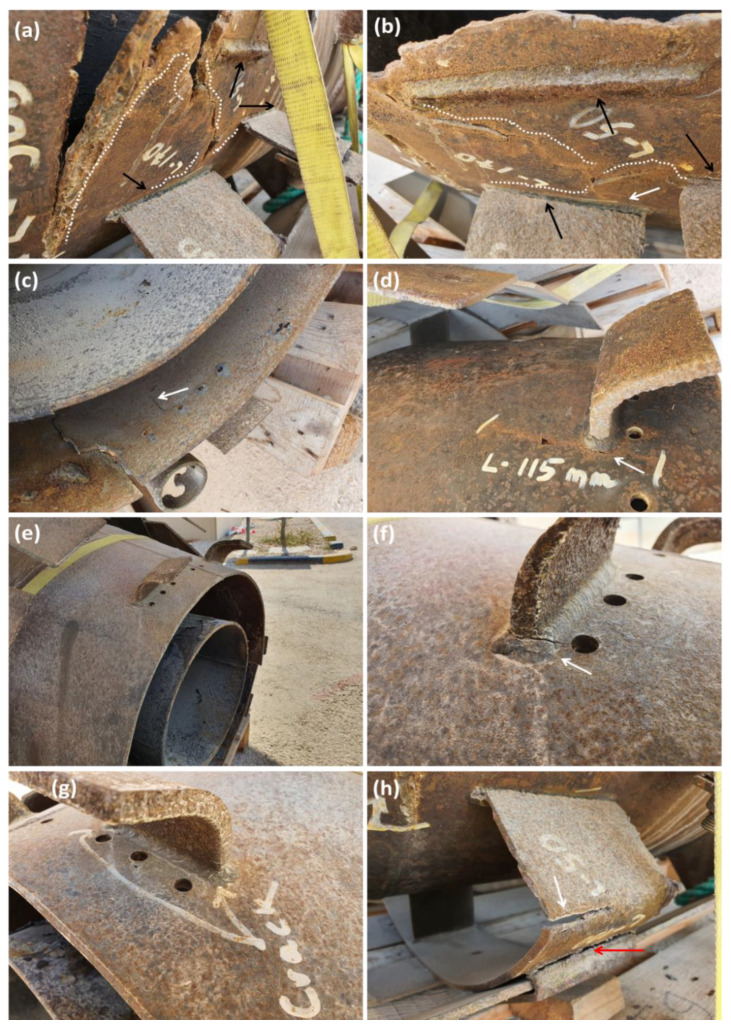
A detailed study of the root cause of the failure region in the vicinity of the joints (**a**–**c**,**f**,**h**) The most critical sections that failure took place (**d**,**g**) welding at the vicinity of holes (**e**) the upper part of the flare tip.

**Table 1 materials-13-03426-t001:** The material specification for different components of the flare.

Row	Name	Material
1	Body Upper	Alloy 800H
2	Body lower	316 L SS
3	HP outlet nozzle	Alloy 800H
4	LP outlet duct	Alloy 800 H
5	Pilot	316 L SS
6	Pilot nozzle	Alloy 550
7	Pilot gas manifold	316 SS
8	Wind deflector	Alloy 800H

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
