# Peer review of "Failure Analysis of a Flare Tip Used in Offshore Production Platform in Qatar"

_materials, 2020, doi:10.3390/ma13153426_

Round 1

Reviewer 1 Report

Dear Authors,

Please take into consideration my recommendations to further improve your paper.

The quality of the figures could be highly improved.

I recommend stating the base materials of the flare tips in the literature overview. For example, the base material of reference 8 is not known in the text.

The sentence „According to the site data, the flare was exposed to a highly aggressive environment, i.e., high temperature 700 ËšC and high humidity, which is typical weather conditions in Qatar.” could be misunderstood.

For the T-joints, which caused the failure, did the WPQR included any testing to examine the HAZ? Such as hardness testing, CVN, etc.

I recommend to connect Table 1. to Figure 1. and mark the different parts in the image, for easier understanding.

Figure captions could be more meaningful (e.g., Figure 3.).

There are several typos in the text, such as: „regent”, „EDA”, etc.

Does Figure 6. shows the unaffected base material? This should also be present in the caption.

I would recommend inserting an image, which shos the original location of the tensile specimen.

Please double-check the numbering of the Figures and their reference in the text.

What are the highlighted areas in Figure 9.? It should be presented in the figure caption.

In Figure 6, what does „corroded” mean? High-temperature oxidation, or sulfurization, etc.?

On page 11 „Another important fact that can be detected after the initial examination of the whole flare structure is that the failure mostly took place in the upper part of the flare body, whereas the lower part showed no failure. It is worth noting that the flare upper body parts are mainly made of Incoloy 800H whilst the lower body part is made of SS316 (Table.1). This clearly indicates that due to the harsh service condition of the flare, including 700 ËšC and 3.8 Bar, the materials selected for upper body parts (Incoloy 800H) might not be chosen appropriately based on the real service condition of the flare.” Does this mean the AISI 316 stainless steel is more applicable for high-temperature service than Incoloy 800H?

In my opinion, the term „T-joint” is not used correctly in the text, as the joints in Figure 11. are not T-joints.

In my opinion, the microstructural examination of the welded joints is essential in this failure analysis, which is exceptionally missing from this paper.

Reviewer 2 Report

Some general comments: 

In my opinion the sub-chapter materials and methods should be rewritten because, as it stands and given what I refer to in my comments, it is not clear where the authors want to start this point.

The labels Fig., Figure and Table should have the same font and number which is not the case.

Some line by line comments:

Page 1 (Abstract) Line 10 “...mechanical characterization, OM and SEM equipped with EDS…” The authors must specify here the initials  Scanning Electron Microscope (SEM) and Optical Microscope (OM) Energy Dispersive Spectroscopy (EDS)Page 3 Line 7 “Table 1 shows the materials used for…” This paragraph should be in the Materials and Methods subtitle.

Page 4 Line 1 “2. Materials and Methods” This subtitle must be rewritten. There are two Fig. 3 (page 4 and 5)

Page 5 Line 3 “...specimens. Fig. 4 shows the…”The Fig. 4.  Is in page 6 Line 20 (?) “3. Results and Discussion 3.1. Microstructural Characterization Fig. 5 shows the chemical composition of…”   Is Fig. 6

Page 8 Line 4 (from the bottom)  “...mechanical properties. Fig.8 demonstrates the tensile…” The figure in page 9 is the Fig. 5. Last line  “...Specimens 1 to 3 represent the tensile behavior of the materials far from the failure, whereas specimens 4 to 5 demonstrate the tensile strength of the samples extracted in the vicinity of the failure….”

This sentence is not in accordance with the figure 8 (5 in the text). There are clearly two types of behaviour: one of the specimens in purple, orange and green (specimens 4, 3 and 5) with high UTS. The other is the specimens in blue and red (specimens 1 and 2) with lower UTS and a strong elongation.

In page 5 the authors said that “Three specimens were prepared for each sample in order to have reliable results”. So, why do they have only 5 specimens in the figure 8 (5 in the text) instead of 6?

Page 9 Line after the figure 8 (5 in the text) “From Fig.8, it can be concluded that Ultimate Tensile Strength (UTS) of the latter, i.e., the materials near the failure parts substantially decrease, up to 44% reduction in UTS. Moreover, the yielding strength also shows a significant reduction from almost 600 MPa to 200 MPa for unaffected and affected materials, respectively.” This sentence is not in accordance with the figure 8.

Page 10 Some comments about Figure 9 I don´t understand the difference between the figure 9 (a) and figure 9 (b). The specimens (1) and (2) of (a) are near or far from the failure, but in (a) or (b)? “...(c-d) SEM image for specimen 1 , (e-f) SEM image for specimen 1…” In the legend of figure 9 there are no differences between (c-d) and (e-f) I suggest that the authors maintain the nomenclature of the specimens referred to in figure 8 (5 in the text). I suggest that the authors put all the figures of the same type of specimens in the same column. For example, in the left column, figures (a), (c) and (d) In the right column, figures (b), (e) and (f)

Line after the figure 9 “...Specimens 1 and 2 represent the specimen near and far from the failure region, respectively. From the macroscopic image (Fig. 9a-b), it is clear that specimen near the failure (specimen 2) manifest more considerable extension, i.e., higher elongation compared with the specimen taken far from the failure, i.e., specimen 2….” The authors in the first and the second line say exactly the opposite. The same in the next two lines. The authors must review the figure 9 and all these sentences about this figure.

Page 11 Figure 6 I suggest that the authors change the type of dots in the figure 6 for better understanding. The authors did not mention the figure 10 (6 in the text) in the text. Line  3,  after  3.3. The Root Cause of Failure“...failure region demonstrated no co change, i.e., showing the…” I don´t understand “...no co change…”

Page 12 Figure 11. Longitudinal and circumference welding line in the flare body part sowing a sound weld. Should be “...circumferential…” and “...showing…”

Page 13 Fig 7. Is the figure 12

Reviewer 3 Report

The manuscript is a technical research report, not a scientific article. Conclusion continues to analyze test results. Contrary to appearances, the manuscript does not meet many requirements set out in the instructions for authors and in the template. Numerous editorial and stylistic errors. The list of shortcomings, irregularities and errors would be long, comparable to the length of work. In my opinion, the manuscript cannot be published, nor can I see any improvement.

Round 2

Reviewer 1 Report

Dear Authors,

Thank you for the responses and the improvement.

Best regards

Reviewer 2 Report

Line 43  “...prevent future immature future failure along…” This sentence is not correct. The word future appears twice.

Line 66 “...Generally, the flare body is generally made up of a rolled sheet, while welding is…” This sentence is not correct. The word generally appears twice.

Line 90 “...was fifteen years. However, it demonstrated immature failure, a maximum lifetime of 15 years.…”This sentence is not correct: fifteen - 15 years??????

Line 97 “..cause of Failure of the flare tip used…” I propose ...failure...

Line 97 “...Accordingly, visualization, chemical analysis, microstructural and mechanical characterizations of samples taken from locations near and far of the observed cracks.” This sentence is not correct. The authors should rewritten this paragraph. 

Line 180 “...Specimens 1a, 1b, and 2a, 2b represent the specimen near and far from the failure region, respectively…”

Line 205 “...Figure 9. Fracture surface morphology : (a-b) macroscopic image for specimen 1 and 2 respectively, (c-d) SEM image for specimen 1 , (e-f) SEM image for specimen 1.  The legend of figure 9 is not in accordance with the sentence in line 180. The legend of (c-d) is equal to (e-f). Again, the sentence in line 181, 182  and 183 are not in accordance with the statement in line 180. The authors must clarify this idea.

Line 262 “...Figure 11. Longitudinal and circumference welding line in the flare body part sowing a sound weld. I suggest “... Longitudinal and circumferential welding... I don´t understand the sentence “...a sound weld….”
